# Deep-Learning-Based Multimodal Emotion Classification for Music Videos

**DOI:** 10.3390/s21144927

**Published:** 2021-07-20

**Authors:** Yagya Raj Pandeya, Bhuwan Bhattarai, Joonwhoan Lee

**Affiliations:** Department of Computer Science and Engineering, Jeonbuk National University, Jeonju-City 54896, Korea; yagyapandeya@gmail.com (Y.R.P.); bhubon240@gmail.com (B.B.)

**Keywords:** channel and filter separable convolution, end-to-end emotion classification, unimodal and multimodal

## Abstract

Music videos contain a great deal of visual and acoustic information. Each information source within a music video influences the emotions conveyed through the audio and video, suggesting that only a multimodal approach is capable of achieving efficient affective computing. This paper presents an affective computing system that relies on music, video, and facial expression cues, making it useful for emotional analysis. We applied the audio–video information exchange and boosting methods to regularize the training process and reduced the computational costs by using a separable convolution strategy. In sum, our empirical findings are as follows: (1) Multimodal representations efficiently capture all acoustic and visual emotional clues included in each music video, (2) the computational cost of each neural network is significantly reduced by factorizing the standard 2D/3D convolution into separate channels and spatiotemporal interactions, and (3) information-sharing methods incorporated into multimodal representations are helpful in guiding individual information flow and boosting overall performance. We tested our findings across several unimodal and multimodal networks against various evaluation metrics and visual analyzers. Our best classifier attained 74% accuracy, an f1-score of 0.73, and an area under the curve score of 0.926.

## 1. Introduction

Emotion is a psycho-physiological response triggered by the conscious or unconscious perception of external stimuli. There is a wide variety of factors associated with emotion, including mood, physical feeling, personality, motivation, and overall quality of life. Emotions play a vital role in decision making, communication, action, and a host of cognitive processes [1]. Music videos are commercial products that pair music with imagery to promote a song or album. Music videos convey affective states through verbal, visual, and acoustic cues. Because they blend multiple types of information, a number of different methods of analysis are needed to understand their contents. In the context of music videos, identifying emotional cues requires not only analysis of sound, but visual information as well, including facial expressions, gestures, and physical reactions to environmental changes (e.g., changes in color scheme, lighting, motion, and camera focus points).

A number of studies have attempted to show how music carries human affective states [2] and boosts mood and self-confidence [3,4,5]. Sometimes, this emotional effect is counterintuitive, as even sad music can evoke pleasure and comfort in listeners [6,7]. Pannese et al. [8] used the conceptual metaphor of a chain of musical emotion that emanates from a recording or performance and works its way to the audience and listeners. The performer or composer transmits emotion at the production level. The music, then, evokes emotion at the level of perception when it is received by the audience or listeners. Finally, an affective state is brought about in the audience or listener in response to the song at the induction level. These authors conceived of emotion using a top-down approach, and were unable to describe psychological changes in affective states when listening to music. The fact that the act of listening and responding to music involves subjective assessments on the part of the listener adds to the complexity and uniqueness of affective computing. Several musical components, including harmony, tonality, rhythm, mode, timing, melody, loudness, vibrato, timbre, pitch, and the vocalist’s performance, make each musical work unique. Visual components further complicate emotional analysis, as performers can add emotions to their music through body movements, gestures, and facial expressions.

With the rise of the Internet and social media, users are engaging more than before with multimedia content in order to communicate and explore their affective states. In this paper, we focused on analyzing music and image sequences produced by either the music industry or random social media users. Music videos are among the most circulated types of content on the Internet. Emotion recognition has already been employed by music streaming services, the video game industry, the advertising industry, the television industry, and the music industry for audio–video content matching and synchronization [9]. Emotion recognition is also used in mental treatment, meditation, and encouragement training [10].

Deep learning technology makes our daily lives more convenient across diversified domains. Numerous success stories have rapidly spread in the domains of animal sounds [11,12,13,14] and music information retrieval [15]. Automatic affective computing of music using deep neural networks (DNNs) allows for the processing of a large number of features from multiple modalities. However, DNNs require a large amount of data for training, and data scarcity has held back research in this field. Other major challenges for emotional analysis in music videos include the undefined frameworks for emotional representation, a lack of standard datasets, and difficulties in annotation due to subjective and affective states that vary across time. Moreover, how individuals demonstrate their emotions varies across cultural groups, languages, and music makers. To compound the problem, multiple information sources (audio, video, facial expressions, and lyrics) are used to communicate information about affective states. Finally, user-generated music videos may not present a consistent picture of emotion across audio and video sources. Annotators, in turn, must consider correlated sources to provide precise annotations.

This article seeks to enhance and improve a supervised music video dataset [16]. The dataset includes diversified music video samples in six emotional categories and is used in various unimodal and multimodal architectures to analyze music, video, and facial expressions. The unimodal term in this paper is used for a network that uses only one source of information, such as music, video, or facial expressions. The integrated structure of more than one unimodal source is termed as multimodal. We conducted an ablation study on unimodal and multimodal architectures from scratch by using a variety of convolution filters. The major contributions of this study are listed below:(a)We extended and improved an existing music video dataset [16] and provided emotional annotation by using multiple annotators of diversified cultures. A detailed description of the dataset and statistical information is provided in Section 3.(b)We trained unimodal and multimodal architectures with music, video, and facial expressions using our supervised data. The networks were designed using 2D and 3D convolution filters. Later, the network complexity was reduced by using a novel channel and separable filter convolution.(c)An ablation study was conducted to find a robust and optimal solution for emotion classification in music videos. Music was found to be the dominant source for emotion-based content, and video and facial expressions were positioned in a secondary role.(d)The slow–fast network strategy [17] was applied for multimodal emotion classification. The slow and fast branches were designed to capture spatiotemporal information from music, video, and facial expression inputs. The learned features of two parallel branches of a slow–fast network were shared and boosted by using a multimodal transfer module (MMTM) [18], which is an extension of “squeeze and excitation” (SE) [19].

We overcame the difficulties of input processing for multimodal data of large dimensions. Different networks were trained and interpreted by analyzing different information sources individually and jointly. The network performance was evaluated visually and statistically by using different evaluators.

The remainder of this article is organized as follows. In Section 2, we present related past studies on music, video, facial expression, and multimodal processing for affective computing. Section 3 explains our music video emotion dataset and the proposed deep neural network architectures. Section 4 includes the statistical and visual test results for the unimodal and multimodal architectures. Finally, Section 5 concludes the study by discussing further directions for future research.

## 2. Related Works

Different approaches to the analysis of emotions in music have been taken in the past by using data-driven explorations. At the time of writing, the authors were not aware of any research on deep learning that leveraged affective computing of music videos from scratch. Some unimodal architectures were studied well by using diverse datasets in the past. In this section, we discuss some existing techniques for affective computing of music, video, and facial expressions.

Music emotion recognition (MER) is one research sub-domain of music information retrieval (MIR) that focuses on the study of the characteristics of music and their correlation with human thinking. The audio can be dominant information for high-level semantics, such as emotion, in multimedia content. Lopes et al. [20] showed the importance of sound in horror movies or video games for the audience’s perception of a tense atmosphere. Particularly for emotions in music, some research has shown satisfactory results by using music audio and lyrics. Naoki et al. [21] integrated their analysis of music lyrics and audio by using a mood trajectory estimation method. Algorithms used to generate these features from audio and to classify them include k-nearest neighbors, random forests, support vector machine (SVM), among other regression models. Song et al. [22] proposed a LastFM dataset classified into four emotional categories (angry, happy, sad, and relaxed) and used an SVM with polynomial and radial basis function kernels for classification. Other studies [23,24,25] used various handcrafted music features for affective computing. The comprehensive work of [26] proposed a dataset and compared various machine learning and deep learning methods. Recurrent neural networks [27,28] and convolutional neural networks (CNNs) [29] generally rely on time–frequency spectral representation for emotion classification in music. Tsunoo et al. [30] used rhythm and bass-line patterns to classify the music contained in the Computer Audition Lab 500-song (CAL500) dataset [31] into five emotional categories. The CAL500 dataset includes 500 popular songs from Western countries with semantic labels derived from human listeners. One CNN-based music emotion classification [32] method for the CAL500 dataset, as well as its enriched version (CAL500exp) [33], was used for the classification of 18 emotion tags in the dataset. Recently, after music source separation [34] and attention [35], individual music sources were also applied to improve prediction of emotions in music by using a spectral representation of audio as the input. The spectrogram was a handcrafted magnitude-only representation without phase information. Orjesek et al. [36] addressed this problem by using a raw waveform input for their classifier. Our study used both the real (magnitude) and imaginary (phase angle) information from audio for emotion classification because several studies [37,38,39] have demonstrated that phase information improves the performance of both speech and music processing.

Video is an ordered sequence of images that include several visual clues for affective computing. While there is no deep-learning-related research on the visual representation of emotion in music videos, several studies [40,41,42] have examined emotion in user-generated videos. The main challenge in the analysis of these videos is the subjective nature of emotion and the sparsity of video frames in which emotion is expressed. By using attention mechanisms, some researchers [43,44] found that it is beneficial to boost the visual cues for emotion in video. In sequential data, temporal information is important. Xu et al. [45] conducted a study on the capture of temporal or motion information by using an optical flow in parallel with the RGB channel. Spatiotemporal data processing is improved by using a slow–fast network [17], where the slow branch carries spatial information with fewer video frames, and the fast branch is responsible for capturing the temporal information with a large number of video frames. Multiple information processing paradigms have also been used on movie clips to capture diverse emotional cues with multimodal representation. Modern music videos use a wide range of filmmaking styles as marketing tools in order to promote the sale of music recordings. Many music videos present specific images and scenes from the song’s lyrics, while others take a more thematic approach. To produce a music video, the director tries to create a visual representation that matches their subjective analysis of the emotion in the piece of music. The classification of emotions in movies by using audio and visual information was proposed in [46,47] with the use of low-level audio–video features. Affective computing of movies was later studied in [48] by using a neural network and face and speech data. Reinforcement learning was used in [49] for identifying funny scenes in movies. These techniques can be useful for affective computing of music videos and dealing with multimedia content.

Facial expressions can provide information about humans’ feelings and thoughts, and they play a crucial role in interaction and decision making. Facial expressions can have universal qualities [50] and have potential applications in human–computer interaction, computer vision, and multimedia entertainment. The facial expressions of a music video actor are crucial for affective computing because he/she is guided by the video director to bring their body movement and expressions in line with the emotions expressed by the music. In relation to other visual cues, such as gestures, background scenery, color, and camera position, facial expressions provide clearer visual cues for emotional analysis [51,52]. Some deep-learning-based research [53,54,55] has achieved satisfying results by using facial expression for different applications. Facial expressions have been extensively studied in speech recognition and have been found to be beneficial for improving learning networks’ capabilities [56,57,58]. Seanglidet et al. [59] proposed the use of facial expression analysis in music therapy; however, facial emotions have not been used in the study of emotions in music videos.

The wide proliferation of multimedia content that is posted online is increasingly pushing researchers away from conventional unimodal architectures and towards complex multimodal architectures. Some multimodal architectures [60,61] have been proposed for affective computing analysis of music videos by using machine learning technology. Pandeya and Lee [16] proposed a supervised music video emotion dataset for a data-driven algorithm and used late feature fusion of audio and video representations after transfer learning. We extended their dataset and incorporated additional emotional cues from music, video, and facial expressions. These information sources treat emotions individually with their own schema of emotion representation. Multimodal representation is one means of dealing with the complex problem of emotional analysis, where each information source within a music video influences the emotions conveyed through the acoustic and visual representation. Affective computing of music videos, however, has not been used to interpret multiple sources of information. Our model is unique inasmuch as it used music, video, and facial expression information. We present various architectures, convolution techniques, and applied information-sharing methods for emotional classification of music videos.

## 3. Data Preparations

A number of emotion representation frameworks have been proposed in the last decade. The categorical model [62] describes human emotions by using several discrete categories. Conversely, the dimensional model [63,64,65] defines emotions as numerical values over two or three psychological dimensions, such as valence, arousal, and dominance. Some variants of these frameworks have been applied in MER studies [34,35,66,67] on music data. Several datasets have previously been proposed for supervised emotional analysis of music. Some of these datasets [68,69,70] follow the categorical model, providing several discrete categories of emotions, and other datasets [24,26,71] used the dimensional model to represent emotions as values in a 2D valence and arousal space. Similarly to those of music, some video emotion datasets [72,73] have been proposed that used the categorical model, and others [74,75] used the dimensional model.

The 2D valence–arousal framework is the most widely used framework for emotion representation in music. While this approach overcomes the problem of categorical limitations and ambiguities in search tags, the categories are vague, unreliable, and not mutually exclusive [76]. The categorical representation can be a better approach for an online streaming service system where the end-user usually makes their demands for their favorite music videos based on class categories, such as emotion tags, singers, or genres. We used the framework identified in [16], where cross-correlation among six basic emotions was explained for music videos, and we extended the dataset. This dataset originally had 720 Excitation, 519 Fear, 599 Neutral, 574 Relaxation, 498 Sad, and 528 Tension data samples. The training, testing, and validation sets were not defined separately, and multiple samples were taken from the same music video. This led to a problem of overfitting, as DNNs are clever in capturing the shortcuts. Shortcuts are decision rules that perform well on standard benchmarks, but fail to transfer to more challenging testing conditions, such as real-world scenarios [77]. In the music video, such shortcuts can be the outer frame of the video, channel logos, and opening or background music. Our updated dataset has nearly twice as much data, which was derived from a wider variety of samples. Most samples were not repeated from a single music video, and the training and testing samples were taken from distinct sources. The statistical layout of our dataset and the number of samples are presented in Table 1.

We categorized the dataset into six distinct classes based on their corresponding emotional adjectives. The “Excited” class usually includes positive emotions. The visual elements of the “Excited” class include images of a smile on a face, movement of arms, dancing scenes, bright lighting, and coloring effects. The audio components of this class include high pitch, large pitch variations, uniform harmony, high volume, low spectral repetition, and diverse articulations, ornamentation, and vibrato. The visual features of the “Fear” class reflect negative emotions via a dark background, unusual appearance, wide eyes, open mouth, a visible pulse on the neck, elbows directed inward, and crossed arms. Common visual elements in the “Tension” class are fast-changing visual scenes, crowded scenes, people facing each other, aggressive facial expressions with large eyes and open mouths, and fast limb movements. The audio elements in the “Tension” and “Fear” classes include high pitch, high tempo, high rhythmic variation, high volume, and a dissonant and complex harmony. The visual elements in the “Sad” class are closed arms, a face buried in one’s hands, hands touching the head, tears in eyes, a single person in a scene, a dark background, and slow-changing scenes. The “Relaxation” class includes ethnic music and is visually represented with natural scenes in slow motion and single-person performances with musical instruments. The acoustic components of the “Sad” and “Relaxation” classes include slow tempo, uniform harmonics, soft music, and low volume. The “Natural” class includes mixed characteristics from all five other classes. The data samples included in each class reflect diversity in music genres, culture and nation of origination, language, number of music sources, and mood. Five coworkers were involved in the construction of the new dataset.

The raw music video data needed to be processed in an acceptable form prior to being entered into the neural network. Our processing of the dataset of music, video, and facial expressions followed a number of steps for each individual data sample. The music network was trained on the real (magnitude) and imaginary (phase angle) components of the log magnitude spectrogram. The magnitude of the log Mel spectrogram was kept to one channel, while the phase angle representation was placed in another in order to preserve both.

For this work, a 30-second audio waveform x_i_ was converted into mono and then subsampled with a window size of 2048, a sampling rate of 22,050 Hz, and a time shift perimeter of 512 samples. The sampling rate varied for the slow–fast network, in which x_i_ in the slow path was sampled at a rate of 32,000 Hz, while the x_i_ in the fast path had a sampling rate of 8000 Hz. Fast-Fourier Transform (FFT) was then applied to each window to transform the x_i_ from a time-domain representation into a time–frequency (T-F) representation (X_i_(t, f)). A total of 128 non-linear Mel scales that matched the human auditory system were selected from across the entire frequency spectrum. The log Mel spectrogram offers two advantages over waveform audio; first, it reduces the amount of data that the neural network needs to process compared with waveform representation, and second, it is correlated with human auditory perception and instrument frequency ranges [78].

The serially binned images were collected in a distributed way to preserve the temporal information of the entire video sequence, as shown in Figure 1. Each video was converted into several-frame sequences, V_i_ = {vτ, v_2_τ, v_3_τ, …, v*_n_*τ}, where τ represents equal time intervals in the video sequence. For each sample, τ changes according to the total number of frames *n* that were extracted. For the video/face network, 64 frames were taken in a distributed fashion. The video data were processed in a similar way to the audio data by varying the frame rate in the slow (eight frames) and fast (64 frames) branches of the slow–fast network.

For the face network input, face areas were detected in each video frame by using the cascade classifier feature of OpenCV (https://opencv.org/ accessed on 7 June 2021) 4.2.0 version. The images were cropped and resized for the network input. Music video frames may or may not have a face in them, or may depict more than one face. We chose music videos that contained at least one face. In each video, video frame v_t_ may contain one face {f_1_}, more than one face {f_1_, f_2_, f_3_, …, f_m_}, or no faces. After processing all of the video frames, we counted all of the frames containing faces. If the number of face frames was less than the face network input size, we repeated these frames until satisfying the requirement. Additional frames were discarded if the total face frames exceeded the neural network input. The repeated frames and discarded additional frames lost the temporal information of video sequences of faces. This is one reason for why our face network had a relatively small contribution to the overall decision making compared to the video and music networks.

After the preprocessing, the input for the audio network was A_N_ = $\overset{N}{\underset{i=0}{\sum }}X_{i}$; for the video network, it was V_N_ = $\overset{N}{\underset{i=0}{\sum }}V_{i}$, and for the face network, it was F_N_ = $\overset{N}{\underset{i=0}{\sum }}F_{i}$, where N is the data used in one batch. The multimodal input was the integrated form of each unimodal input.

## 4. Proposed Approach

### 4.1. Convolution Filter

Several networks were designed and integrated in a numbered way to find the optimal structure. The general 2D and 3D convolution filters were used in the music and video/face networks, respectively. Particularly in video processing, 3D convolution has been found to be better in capturing the spatial and motion information, but it exponentially increases the system’s complexity. Popular 3D networks [79,80] have a great complexity and, hence, require large amounts of data for successful training.

In this paper, the complexity of 3D convolution was reduced by using separable filter and channel convolution. For the separable filter, the 3D convolution filter of size n×n×n was divided into 2D space as 1×n×n and n×n×1. As illustrated in the right-most column of Figure 2, the 3D convolution filter was split into a 2D space filter with channel separation. The proposed convolution was an integrated form of separable channel convolution [81] (second column) and (2 + 1)D convolution [82] (third column). The idea of the separable filter and channel convolution was also used for the 2D audio network. The square filter of the 2D convolution was divided into a temporal filter (1×n) and a spatial filter (n×1), as in [83]. The channel size was reduced to one in the sequential block of the dense residual network for the separable channels. By using a novel separable channel and filter convolution, we drastically reduced the complexity and improved the system’s performance for both the music and video networks.

### 4.2. Proposed Networks

We propose four basic architectures for the music, video, and face emotion networks. The architecture A_0_ in Figure 3 only used 2D/3D convolution, and the other architectures (A_1_ to A_3_), which are shown in Figure 4, were designed with a separable 2D/3D channel and filter convolution. A_3_ was a slow–fast network designed to capture the spatial and temporal information of audio/video. The basic architecture of our proposed network and a detailed view of each block are shown in Figure 3 and Figure 4. In addition to the proposed networks, the well-known C3D network [79], which was trained on the sport-1M dataset [84], was also used for video and facial expression recognition. The numbers of filters in each of the five convolution layers (1–5) were 64, 128, 256, 512, and 512, respectively. The dimensions of the input network (height, width, channel, number of frames or depth) were equal to 112, 112, 3, and 32. The original C3D network was modified in a bottleneck layer with a dropout value of 0.2. The modified C3D network was the same as the one in [16], which helped us to make a fair comparison with this study based on the parameters and evaluation scores. For a detailed look at the architecture, the reader is invited to refer to the original paper [79]. The pre-trained network for music video emotion classification was not found to be beneficial in terms of performance and complexity.

The basic block of the unimodal architecture, which is shown in Figure 3 and Figure 4, was further integrated for the multimodal representations. The feature information from multiple branches was merged to enhance top-level decision processing. A review [85] illustrated early, late, and mid-level fusion in multimodal affective computing. Differently from this research, our network learned information and integrated it after each block of the dense residual network by using MMTM. The MMTM and SE networks were used for information sharing and boosting during training. In the ablation study, we performed an analysis of these blocks with several unimodal and multimodal architectures. At the decision level, all of the branch information was globally aggregated and computed for the class-wise probability. The Softmax function was used in the final layer of the neural network, which mapped the output nodes in a probability value range between 0 and 1. We used the categorical cross-entropy loss function for a one-hot vector target. This function was used to separately compute the binary cross-entropy for each class and then sum them up for the complete loss.

## 5. Experimental Results

To make this report understandable and comparable with those of different research groups, we consistently report our experimental data. Several networks are compared based on the evaluation score, complexity, and visual analysis by using a confusion matrix and a receiver operating characteristic (ROC) curve. We define accuracy as the probability of correct classification within a dataset. Accuracy indicates better evaluations in a balanced dataset. The F-measure is the harmonic mean of precision and recall; when the precision increases, the recall decreases, and vice versa. The F-score handles imbalanced data and provides a measure of the classifier’s performance across different significance levels. A confusion matrix presents the number of correct and erroneous classifications by specifying erroneously categorized classes. A confusion matrix is a good option for reporting results in multi-class classification. The area under the ROC curve is a measure of how well a parameter can distinguish between a true positive and a true negative. An ROC is a probability curve that provides a measure of a classifier’s performance in two-dimensional space. The area under the ROC curve (AUC) measures the degree of separability across multiple classification thresholds. These evaluation metrics were used to evaluate the effectiveness of our system for emotional classification in music videos. The data (https://zenodo.org/record/4542796#.YCxqhWgzaUk accessed on 7 June 2021) and code (https://github.com/yagyapandeya/Supervised-Music-Video-Emotion-Classification accessed on 7 June 2021) used to produce these experiments are publicly available (in Appendix A).

### 5.1. Results of the Unimodal Architectures

The unimodal architectures for music, video, and facial expressions were separately trained and tested. The testing dataset included 300 music video samples that were never used in the training process. These samples were equally distributed in six musical categories for a fair comparison. To measure the performance in terms of our evaluation metrics, the respective ground truth was provided for each test sample. An ablation study was performed to find an optimal network architecture that used both the unimodal and multimodal architectures. The system performance varied with the networks that used SE and MMTM. The SE network was proven to boost system performance. We only used the SE block in our unimodal architecture. In the A_0_ unimodal architecture, the SE block proved to be beneficial. In the other networks, however, it was not found to be effective. The separation in the channel and convolution filter diversified the focal points of the network. The MMTM is an extended form of the SE block that allows more than one modality to share and enhance the information that it learns. The A_3_ unimodal architectures were tested with the MMTM, and the results are illustrated in Table 2, Table 3 and Table 4. In the A_3_ music architecture, the slow and fast paths did not properly synchronize due to their different sampling rates. Therefore, the music networks with or without MMTM showed poor performance. The A_3_ face network showed poor performance because the temporal patterns of the face/video sequence were lost due to repeated or discarded frames. The MMTM block was found to be useful in the case of the A_3_ video architecture. In this architecture, the temporal information was preserved and synchronized with spatial information during training. Each unimodal architecture was evaluated here in relation to multiple networks. The effects of the MMTM and SE blocks were evaluated for each network. These blocks placed increased complexity on the system, but were found to be more efficient in some cases.

We evaluated the emotions in facial expressions based on the facial information of the music video actors. The face network performed poorly in relation to the music and video network because it could not deal with video frames with no faces or multiple faces. Uncertainty appeared in the system when it was presented with faces from an audience or supporting actors. Additionally, the system could not comprehend faces that were blurry and that were presented with low resolution, and this confusion reduced the performance. The emotional cues of the face, however, were still found to be important because they could boost the overall system’s decisions. Table 2 shows the evaluation scores of the various face network architectures that we tested.

The musical analysis focused on the objective aspect of musicality. The neural network determined the changes in the spectral representations according to the emotional category. The unimodal architecture, which used only music information, performed the best compared to the face and video networks. The success of the network was related to the smooth changes in the musical patterns over time. Uncertainty in the music processing network, however, could arise due to the subjectivity of musical components and expressive techniques. We tested various music network architectures for emotion in music, and they are illustrated in Table 3. The A_3_ music networks with and without MMTM had low performance rates because the spectral representation with varying sampling rates could not be synchronized. The single-branch network (2D, slow and fast) performed better with fewer parameters. Both positive and negative effects were found when using the squeeze-and-excitation blocks with the music networks.

The results of the video network were better than those of the face network, but not as good as those of the music network. The video network had a smooth temporal pattern that could not occur in the face network because a non-face frame would break the sequence. Compared to the music, the visual scenes abruptly changed according to time, which could affect the system’s performance. Uncertainties in the video network could occur with user-generated videos, which may not have industry standards of recording, camera movement, and focus. We used the slow–fast network architecture (A_3_) to capture the spatial and temporal information of videos with varying frame rates on each branch. The learned information of each branch was boosted and shared by using the MMTM block. Table 4 reports the various architectures and their scores on the evaluation metrics. Even though it had relatively large training parameters, the slow–fast network with MMTM performed the best when compared to the performance of the other architectures.

### 5.2. Result of the Multimodal Architecture

This study integrated several unimodal architectures for an efficient and optimal solution for emotion classification in music videos. The multimodal architecture was designed in two ways: using music and video information and using music, video, and facial information. We tested several combinations and obtained effective results.

Several combinations of music and video networks are possible; the best-performing multimodal architectures are shown in Table 5. While multimodal architectures that use the face network with audio or video were a possible network option, the contributions of the face network were minimal, and hence, the results are not discussed. The video and music architectures were found to be the dominant sources for overall prediction. With the multimodal video architecture, the MMTM block guided the two parallel branches of the slow–fast network by maintaining the proper synchronization of the learned information. The A_0_ audio network outperformed the others in integration with video and achieved the same accuracy as our best multimodal architecture that used music, video, and facial information, as shown in Table 6.

The integrated network of music, video, and facial expressions was trained in an end-to-end manner by using supervised data. Table 6 reports the integrated results for music, video, and facial expression information. The multimodal architecture that only used 3D convolution has extensive parameters, but the performance did not exceed that of the best unimodal music or video architecture. The slow–fast video network (A_3_) with MMTM performed the best with the integrated architecture. The music network with 2D convolution (A_0_) was found to be better than the network with the rectangular filter (A_1_ and A_2_). The first row of Table 6 shows our top-performing networks. The multimodal architecture using the A_2_ network with music, video, and facial expressions (eighth row) had the lowest number of parameters, but the performance was even lower when using the best unimodal music architecture. The A_2_ face network with SE was found to be effective in the integrated architecture with both music and video networks. For example, the A_2_ face network with the A_1_ music and video network (last row of Table 6) performed better than the A_1_ face network (penultimate row). In the overall analysis, the visual clues that used face expression were found to be supportive for the classifier, with a small increment in network complexity.

### 5.3. Analysis Based on Visual Predictions

We validated the results of our experiments by using two visual evaluation methods: a confusion matrix and a ROC curve plot. The confusion matrix counted the number of samples in classes that were confused with each other. The confusion matrix in Figure 5 shows that the “Neutral” class was highly confusing for our classifier because it held data that were similar to those of more than one class. The classifier result showed confusion on the samples from the “Fear” and “Tension” classes because both classes held similar music structures (mostly rock and metal music). The rock and metal music samples also had common visual characteristics, such as angry facial expressions, dark backgrounds, and unique gestures and appearances. This number of commonalities confused the classifier. The “Sad” and “Relaxation” music videos had a similarly silent nature, so the classifier also confused these classes.

The ROC curves obtained when using our various multimodal architectures are shown in Figure 6. The multimodal architecture with music, video, and facial expression information performed the best. All of the classifiers showed similar ROC curves, with a small difference in the area under the curve (AUC) scores. Networks with similar performance and relatively fewer parameters obtained higher ROC-AUC scores.

Human emotions can be connected to each other, and these connections also appear in music videos. We analyzed the correlation of our six emotional categories in music videos. The class-wise probability was computed by using the sigmoid function at the end of the neural network. The class correlation results of test samples from each class are illustrated in Table 7. A single frame from each sample is provided for illustration. The results show that “Neutral” class carried various common features of the other remaining emotional classes. In addition, the samples from the “Fear” and “Tension” classes were found to be correlated with each other in this experiment, while the samples from the “Excited” and “Sad” classes were not found to be correlated.

An emotional relationship between consumers and music videos was shown in [86,87,88] to be cross-culturally essential for promoting a song or album. We present the results of our best-performing network on two music video with billions of views on YouTube at the time of writing. The prediction results for *Despacito* (https://www.youtube.com/watch?v=kJQP7kiw5Fk) (accessed on 7 June 2021) (7.41B views) and *Gangnam Style* (https://www.youtube.com/watch?v=9bZkp7q19f0) (accessed on 7 June 2021) (4.10B views) are shown in Figure 7 on the left and right, respectively. The prediction results (in percentages) for each class are illustrated with individual curves depicted across 30-second increments. We used the sigmoid activation function to find the correlation of our six emotional categories. The highest-activated class is illustrated with the highest score, and the second or third most probable classes are further down on the vertical axis. One video frame is illustrated at the bottom according to the time for illustration, but the musical components were also responsible for the final decision.

### 5.4. Comparisons with Past Studies

Our research proposes a new deep learning method and prepares a new dataset for emotion classification in music videos. The past studies were conducted for affective computing on low- and mid-level audio and video feature classification using conventional classifiers, such as SVM. No study has implemented a deep learning method that can collectively use audio, video, and facial expression information in a single network with end-to-end training.

A previous study [16] attempted to use a similar music video dataset and a multimodal architecture (MVMM) to analyze music and video. The study implemented a two-stage process in which audio and video features were classified after transfer learning. The model resulted in a good evaluation score, as their test samples were taken from similar training samples. However, the pre-trained C3D network used in [16] had a relatively large number of parameters to which even more parameters were added by our combination of this network with other audio networks (a more detailed comparison of the performance and complexity of our network with those of the C3D network appears in Section 5). Our model reduced the network complexity and was evaluated with a more diverse set of test samples.

Some studies have proposed the use of a diverse dataset for affective computing of music videos. One study concerning emotional analysis of music that used a recurrent neural network with an SVM on top [28] achieved a classification accuracy of 0.542 with the LastFM dataset. CNN-based music emotion classification [32] achieved the highest F1-scores when using the CAL500 (0.534) and CAL500 exp (0.709) datasets with 18 emotion tags. The authors of [25] used an SVM for low-level feature classification of music, and ultimately attained the highest F1-score of 0.764. The results of this study were based on this research team’s self-developed dataset, which consisted of 900 audio clips and associated subjective annotations that were applied consistently with Russell’s emotion quadrants.

In Table 8, we quantitatively compare the current study with past related research. The proposed classifier could not outperform the quantitative results, but they were qualitatively robust because the networks were trained on a relatively large data sample with three sources of input. Hence, the network’s capabilities were more diverse and applicable for real-world applications. The results of the visual analysis support this claim.

A number of possible multimodal architectures that use music-related audio and video information have also been studied for affective computing. The study in [66] used low-level video features (lighting key, shot boundary, color, motion) and audio features (zero crossing, MFCC, delta MFCC) for emotion classification by using Gaussian mixture model (GMM) classifiers. This research team used the DEAP120 dataset with an eight-class category and reached 90% accuracy. The team behind [60] used 140 annotated music video samples from the CAL500 dataset in a model that had the highest level of accuracy when used on audio and video with additional optical flow features (74.4%) or with audio and ImageNet features, as well as the Calibrated Label Ranking (CLR) classifier (72.24%). Two additional studies [60,66] were performed with a high accuracy despite the limited number of data samples that they were tested against. In each of these cases, emotion annotation was provided by the video or audio audience, a fact that differentiates our labeled video clips. Moreover, these methods used conventional emotion classification methods, unlike our end-to-end deep neural network, which accounts for multimodal information.

In this research, first, we analyzed the unimodal representations of music, video, and facial expression information from music videos by using end-to-end training. The unimodal architectures were further integrated into a multimodal architecture to develop a robust and optimal classifier. Compared to past research, our dataset is more diversified; the networks were trained and tested on real data domains and not on features. We reduced the system’s complexity and enhanced the performance of the architecture by using novel convolution and information-boosting methods. The results were statistically evaluated by using various evaluation metrics.

## 6. Conclusions

Affective computing enables AI systems to interpret human emotions. This area of computing is inherently interdisciplinary, though the analysis of emotions in music videos remains a particularly unexplored area within the field of computer engineering. Our system classifies music videos by using a dataset that was introduced for supervised training. Several unimodal and multimodal architectures have been proposed to analyze music, video, and facial expressions from scratch. Our proposed architectures, including the slow–fast network, were designed to use 2D and 3D convolution, as well as a novel separable channel and filter convolution. Our best multimodal architecture achieved 74.00% accuracy, an F1-score of 0.73, and an area under the curve (AUC) score of 0.923.

Future researchers have space for improvement in terms of the performance, the dataset, the emotion representation framework, and the evaluation measures. In this study, we only included music, video, and facial expression networks. Music lyrics are another vital source of information that can be integrated in further studies for more accurate affective computing.

## Figures and Tables

**Figure 1 sensors-21-04927-f001:**
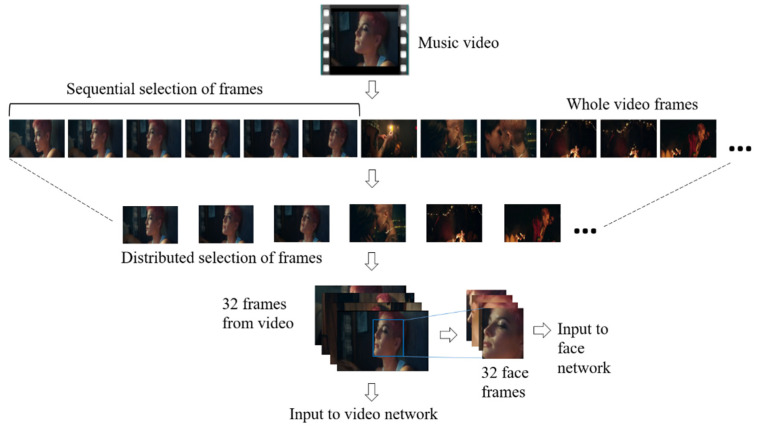
Input video processing by using the distributed selection of frames and faces.

**Figure 2 sensors-21-04927-f002:**
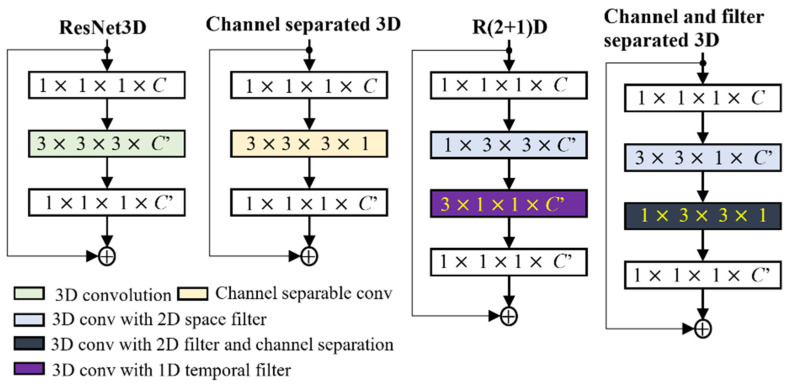
The 3D convolution and its variants in the residual block representation.

**Figure 3 sensors-21-04927-f003:**
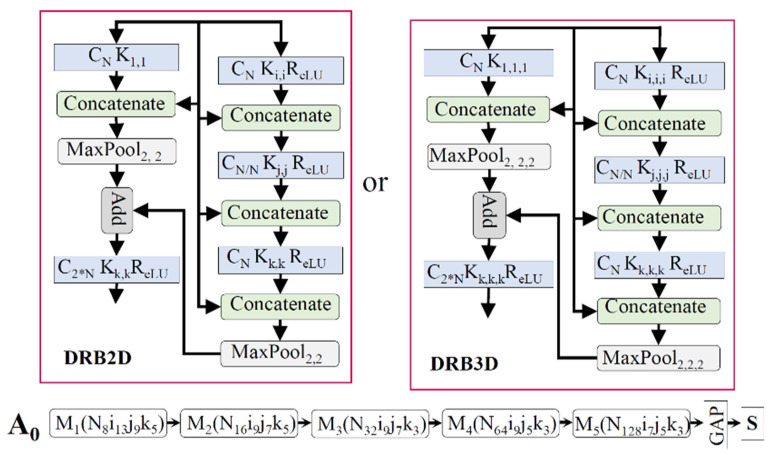
The block diagram of the proposed music video emotion classification network A_0_ with the 2D/3D convolution. The acronyms DRB2DSC and DRB3DSC are detailed views of the dense residual block with the standard 2D and 3D convolution for the music network and the video/face network, respectively. The symbol A represents the network architecture, M represents the dense residual block defined in the detailed view, S represents the Softmax function, GAP is global average pooling, and MMTM is the multimodal transfer module. Similarly, the symbols N, i, j, and k with values in the lower case represent the values of the items, as illustrated in the respective detailed views.

**Figure 4 sensors-21-04927-f004:**
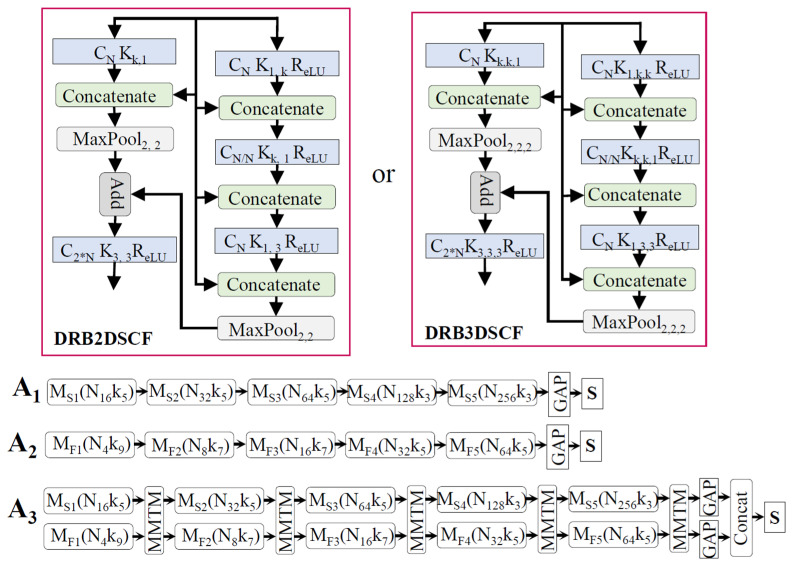
The block diagram of the proposed music video emotion classification network using slow (A_1_), fast (A_2_), or slow–fast network with MMTM (A_3_) and with the separable channel and filter convolution. The acronyms DRB2DSCFC and DRB3DSCFC are the detailed views of the dense residual block with 2D and 3D separable channel and filter convolution for the music and video/face networks. The symbols have the same meanings as the symbols in Figure 3.

**Figure 5 sensors-21-04927-f005:**
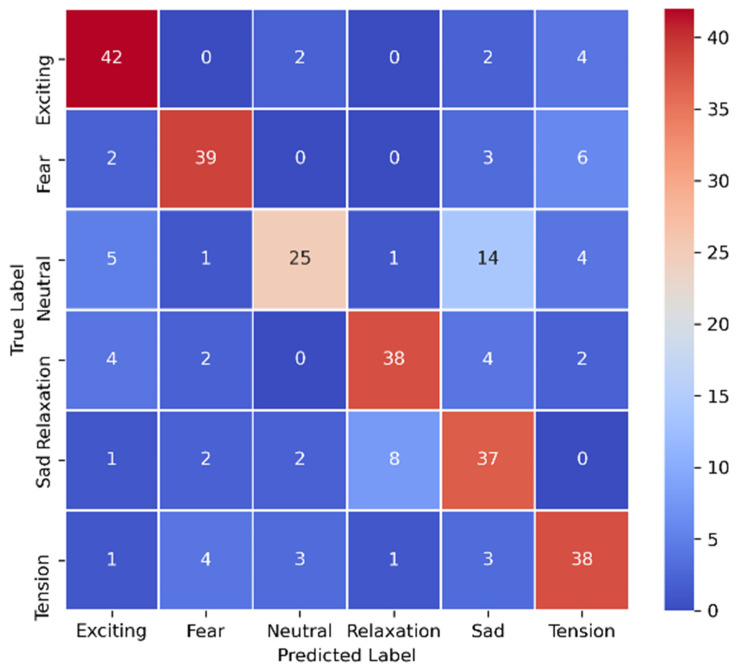
The confusion matrix using our best-performing multimodal architecture.

**Figure 6 sensors-21-04927-f006:**
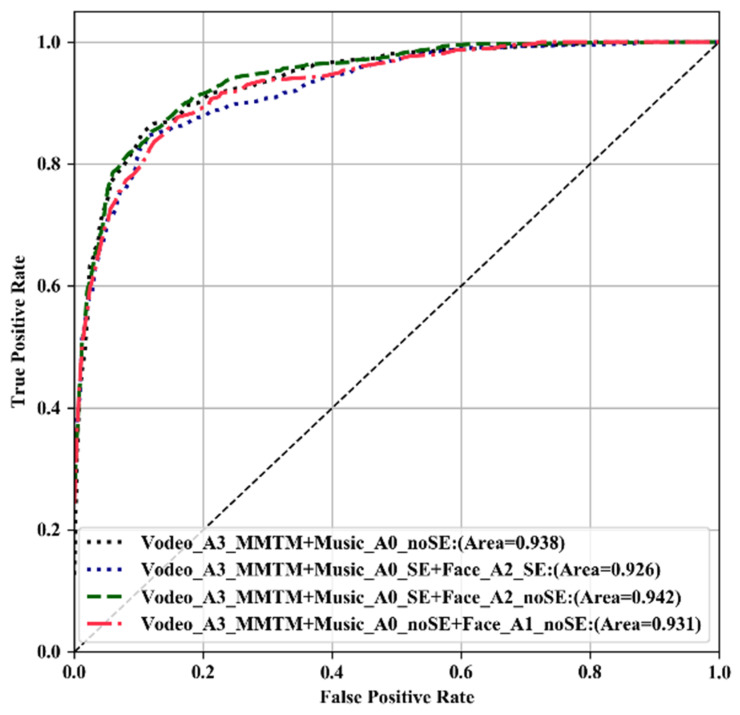
The ROC curve using several architectures for music, video and facial expressions.

**Figure 7 sensors-21-04927-f007:**
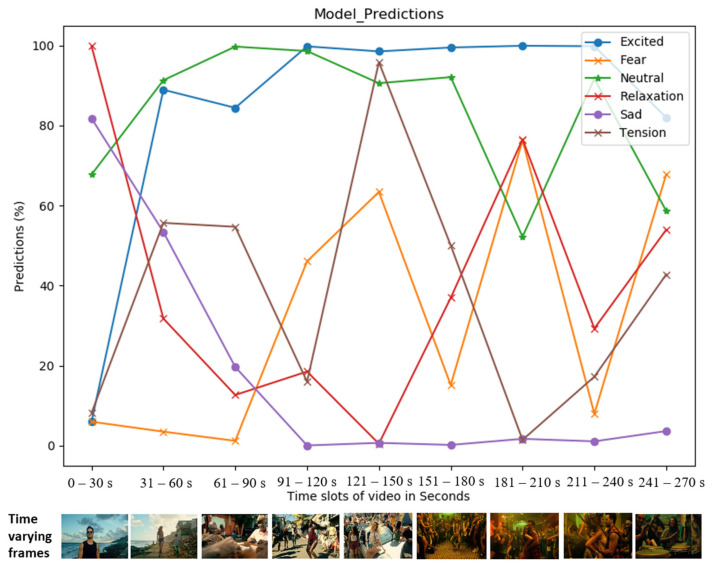
Time-synchronized emotional predictions for specific music videos. Prediction results for the *Luis_Fonsi* (**top**) and *Gangnam Style* (**bottom**) music videos.

**Table 1 sensors-21-04927-t001:** Music video dataset with various adjectives and statistics in each emotion class.

Emotion Class	Emotion Adjectives	Training Samples	Validation Samples	Testing Samples
Excited	Happy, Fun, Love, Sexy, Joy, Pleasure, Exciting, Adorable, Cheerful, Surprising, Interest	843	102	50
Fear	Horror, Fear, Scary, Disgust, Terror	828	111	50
Neutral	Towards (Sad, Fearful, Exciting, Relax) Ecstasy, Mellow	678	99	50
Relaxation	Calm, Chill, Relaxing	1057	148	50
Sad	Hate, Depressing, Melancholic, Sentimental, Shameful, Distress, Anguish	730	111	50
Tension	Anger, Hate, Rage	652	84	50
Total		4788	655	300

**Table 2 sensors-21-04927-t002:** Test results using facial expression.

Model	Test Accuracy	F1-Score	ROC AUC Score	Parameters
C3D	0.3866	0.39	0.731	57,544,966
A_0_ without SE (Face_A_0__noSE)	0.460	0.45	0.778	19,397,078
A_0_ with SE (Face_A0_SE)	**0.516**	**0.51**	**0.820**	19,409,478
A_1_ without SE (Face_A_1__noSE)	0.4933	0.46	0.810	11,876,982
A_1_ with SE (Face_A_1__SE)	0.490	0.46	0.794	11,924,566
A_2_ without SE (Face_A_2__noSE)	0.430	0.37	0.769	**845,670**
A_2_ with SE (Face_A_2__SE)	0.403	0.37	0.755	849,406
A_3_ without MMTM (Face_A_3__noMMTM)	0.449	0.42	0.781	24,083,846
A_3_ with MMTM (Face_A_3__MMTM)	0.419	0.41	0.782	24,174,918

The bold number represents the highest evaluation score and lightweight network architecture (rightmost column).

**Table 3 sensors-21-04927-t003:** Test results using music information.

Model	Test Accuracy	F1-Score	ROC AUC Score	Parameters
A_0_ without SE (Music_A_0__noSE)	0.5900	0.58	0.863	3,637,142
A_0_ with SE (Music_A_0__SE)	0.5766	0.61	0.852	3,659,782
A_1_ without SE (Music_A_1__noSE)	0.5366	0.51	0.859	3,946,949
A_1_ with SE (Music_A_1__SE)	**0.6466**	**0.62**	**0.890**	3,994,533
A_2_ without SE (Music_A_2__noSE)	0.6399	0.61	0.897	**261,297**
A_2_ with SE (Music_A_2__SE)	0.6266	0.61	0.878	267,369
A_3_ without MMTM (Music_A_3__noMMTM)	0.3166	0.22	0.635	4,208,240
A_3_ with MMTM (Music_A_3__MMTM)	0.2433	0.17	0.610	7,941,004

The bold number represents the highest evaluation score and lightweight network architecture (rightmost column).

**Table 4 sensors-21-04927-t004:** Test results using video information.

Model	Test Accuracy	F1-Score	ROC AUC Score	Parameters
C3D	0.3266	0.19	0.723	57,544,966
A_0_ without SE (Video_A_0__noSE)	0.4233	0.36	0.742	19,397,078
A_0_ with SE (Video_A_0__SE)	0.4833	0.46	0.806	19,409,478
A_1_ without SE (Video_A_1__noSE)	0.4099	0.39	0.754	11,876,982
A_1_ with SE (Video_A_1__SE)	0.3666	0.35	0.736	11,922,518
A_2_ without SE (Video_A_2__noSE)	0.3633	0.33	0.710	**845,670**
A_2_ with SE (Video_A_2__SE)	0.3866	0.34	0.727	849,406
A_3_ without MMTM (Vodeo_A_3__noMMTM)	0.4666	0.44	0.774	12,722,646
A_3_ with MMTM (Vodeo_A_3__MMTM)	**0.5233**	**0.53**	**0.837**	24,174,918

The bold number represents the highest evaluation score and lightweight network architecture (rightmost column).

**Table 5 sensors-21-04927-t005:** Test results using music and video information.

Model	Test Accuracy	F1-Score	ROC AUC Score	Parameters
Vodeo_A_3__MMTM + Music_A_0__noSE	**0.7400**	**0.71**	**0.938**	27,812,054
Vodeo_A_3__MMTM + Music_A_1__noSE	0.6733	0.66	0.919	28,121,861
Vodeo_A_3__MMTM + Music_A_2__noSE	0.6399	0.64	0.896	**24,436,209**

The bold number represents the highest evaluation score and lightweight network architecture (rightmost column).

**Table 6 sensors-21-04927-t006:** Integrated test results using music, video, and facial expressions.

Model	Test Accuracy	F1-Score	ROC AUC Score	Parameters
Vodeo_A_3__MMTM + Music_A_0__SE + Face_A_2__SE	**0.74000**	**0.73**	0.926	28,660,878
Vodeo_A_3__MMTM + Music_A_0__SE + Face_A_2__noSE	0.73333	0.72	**0.942**	28,589,478
Vodeo_A_3__MMTM + Music_A_0__noSE + Face_A_2__noSE	0.73333	0.71	0.939	28,657,718
Vodeo_A_3__MMTM + Music_A_0__SE + Face_A_1__noSE	0.6899	0.69	0.917	39,369,624
Vodeo_A_3__MMTM + Music_A_0__noSE + Face_A_1__noSE	0.71666	0.71	0.931	39,689,030
Video_A_1__noSE + Music_A_0__noSE + Face_A_2__noSE	0.69666	0.70	0.912	16,356,198
Video_A_2__noSE + Music_A_0__noSE + Face_A_2__noSE	0.68666	0.67	0.915	4,587,350
Video_A_2__noSE + Music_A_2__noSE + Face_A_2__noSE	0.610	0.59	0.873	**1,433,649**
Video_A_1__noSE + Music_A_1__noSE + Face_A_1__noSE	0.63666	0.63	0.860	19,432,869
Video_A_1__noSE + Music_A_1__noSE + Face_A_2__noSE	0.69999	0.69	0.925	11,942,805

The bold number represents the highest evaluation score and lightweight network architecture (rightmost column).

**Table 7 sensors-21-04927-t007:** Test results using the best multimodal architecture.

Video Frame	Class-Wise Probability	Video Frame	Class-Wise Probability
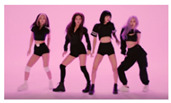 Ground truth: *Excited*	Excited: 0.999981 Relax: 0.884070 Fear: 0.34791592 Neutral: 0.251401 Sad: 0.008204 Tension: 0.006412	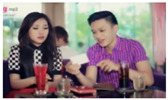 Ground truth: *Neutral*	Neutral: 0.999665 Sad: 0.373862 Relax: 0.301067 Tension: 0.287840 Excited: 0.109657 Fear: 0.004261
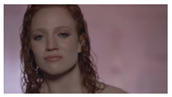 Ground truth: *Sad*	Sad: 0.998427 Relax: 0.741548 Neutral: 0.402856 Tension: 0.294600 Excited: 0.027875 Fear: 0.004099	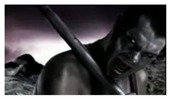 Ground truth: *Fear*	Fear: 0.998849 Tension: 0.945754 Excited: 0.593985 Neutral: 0.374574 Sad: 0.003163 Relax: 0.002293
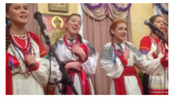 Ground truth: *Relaxation*	Relax: 0.980475 Excited: 0.973017 Tension: 0.397293 Neutral: 0.220959 Sad: 0.177647 Fear: 0.014223	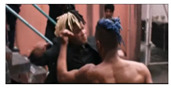 Ground truth: *Tension*	Tension: 0.999754 Neutral: 0.777341 Fear: 0.4758184 Sad: 0.03204632 Relax: 0.03029701 Excited: 0.008435

**Table 8 sensors-21-04927-t008:** Comparison with past studies.

Method	Dataset	Data Type	Emotion Class	Score
RNN [25]	LastFM	Music	4	0.542 (Accuracy)
CNN [22]	CAL500	Music	18	0.534 (F1-score)
	CAL500 exp	Music	18	0.709 (F1-Score)
SVM [21]	Own	Music	4	0.764(F1-Score)
GMM [56]	DEAP120	Music and video	8	0.90 (Accuracy)
CLR [50]	CAL500	Music and video	18	0.744 (Accuracy)
MM [50]	Own	Music and video	6	0.88 (F1-Score)
Our	Own	Music and video	6	0.71 (F1-Score)
Our	Own	Music, video and Face	6	0.73 (F1-Score)

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
