# Peer review of "Deep-Learning-Based Multimodal Emotion Classification for Music Videos"

_sensors, 2021, doi:10.3390/s21144927_

Round 1

Reviewer 1 Report

I found the paper interesting and relevant. However, I would like to point some issues that must be improved:

  • Abstract: when saying that the proposed approach is more precise and rigorous what does it mean? Does it mean that the precision is better? how much? What does it mean more rigorous? I think clear quantification of improvement must be made. And the same is applied in the conclusion.
  • Page 2 (lines 84-85): when it says the dataset is extended nearly twice, please, specify how many samples are added in order to quantify them.
  • A glossary section should be added because there are many acronyms in the manuscript. In addition, the first time an acronym appears should be explained (see DRB2D, DRB3DD, etc.)
  • Page 9 line 376. I have checked the github repository and I do not find the code. I see audio, hdf5 and video files, but no code. It should be added and described in the README file of Github to be easily found.
  • Tables 2-6. When you put parameters in the table, to what parameters do you refer to? Do you mean 1 parameter or more than one?
  • Table 7: The fourth column what does it indicate? And the prediction column (1st column) what is it? accuracy?

Author Response

Dear Reviewer,

Thank you for reviewing and giving us the opportunity to manuscript entitled "Deep Learning-Based Multimodal Emotion Classification for Music Videos" (ID: sensors-1278832). We are glad to submit a revised version of our manuscript for consideration at the Sensors.

We appreciate the time and effort that you have devoted to providing your helpful feedback on our manuscript. We are grateful to you for your insightful and productive comments. The suggestions have made this a much better paper.

We have carefully considered and addressed the comments and integrated changes to indicate the comments raised by the reviewers. We have also extensively rewritten the text according to the reviewers' suggestion to make it less speculative and highlighted the changes within the manuscript. All the authors have equally contributed towards the revision of the manuscript. Please see below for a point-by-point response to your valuable comments and concerns. Our responses are shown in Blue italics, while reviewers' comments are in Black Bold italics. We look forward to your response and hope the revision will enable you to accept this version of the manuscript.

Please find the detail of our response to your valuable comments in the attached file. 

We look forward to hearing from you and thank you for your valuable and constructive comments.

Sincerely,

The Authors

Reviewer 2 Report

Although the research presented by the authors is interesting and decently presented, in my opinion it does not fit the scope of either the special issue, the section, or the journal. This paper focuses on emotion recognition from music videos. The presented approach does not use any sensors, unless we take a regular video camera as one. 

While the method itself, described in Section 4, is promising and looks worthwhile, the construction of the entire article needs to be significantly modified.

The abstract should be rewritten. It does not show the novelty of the study conducted. It is not clear whether the goal is to extend and improve existing music video dataset or to propose a new emotion recognition method based on such dataset. 

The introduction is poorly composed. There is a lack of clearly stated purpose and motivation for the study. While reading it, I got the impression that the goal was to prepare a dataset, while the rest of the article focused on a method for recognizing emotions from music videos. Additionally, this section is full of inaccuracies that make it difficult to read, such as: 

  • Line 36. Who are we? We, the authors, or we as humans? 
  • Line 48. Does the "The study" refer to Pannese et al. work, or to the authors' contribution. If the former, does the entire paragraph refer to this paper?
  • Line 58. "This experiment" - what experiment? 
  • Line 66. How this paragraph connects with the previous part of the article? I guess that it's some kind of motivation part, if so it should be clearly stated. 

The authors claim to have refined the music video dataset they prepared in a previous study. However, a description of this process is lacking. How were the new clips selected? By how many annotators were they labeled?

Related work section focuses on components of  emotion recognition methods that are later used by the authors in their proposal. However, a comparison with other approaches to emotion detection of video clips is missing. Furthermore, this section misses description of datasets used in 5.4 section (LastFM or CAL500*).

If I understand correctly, the proposed dataset consists of emotionally labeled video clips. However, in Section 5.1, the results of the model trained on this dataset are compared with models using, among others, the DEAP dataset, which consists of emotion labels of video viewers. As the authors themselves note, the emotions of the viewer are not always the same as the emotions depicted in the video. Therefore, such a comparison makes no sense. 

I encourage the authors to incorporate the changes and prepare a new version for a more appropriate venue. 

Author Response

(The authors gave the same response as above.)

Reviewer 3 Report

The paper proposes multimodal analysis of music using emotion recognition and facial expression. The performance of this approach has been assessed using various evaluation metrics. It has been shown that it is more precise and rigorous.

The paper is well organized and written. It will be of interest to the journal audience. I do not have any critical comments, except some technical remarks about the figures. It seems Figure 6 is missing. The text in Figures 5 and 7 is too small - it is impossible to read. 

Author Response

(The authors gave the same response as above.)

Round 2

Reviewer 1 Report

I would like to see the code in the Github repository. In the paper, it is stated "The data and code to produce these experiments is publicly available."

All my other concerns are fixed.

Author Response

Dear Editor,

Thank you and the reviewers for giving us the opportunity to manuscript entitled "Music Video Emotion Classification Using Slow-fast Audio-video Network and Unsupervised Feature Representation" (ID: af80e490-a5a6-4ec0-9efb-bfc4d5e608ab). We are glad to submit a revised version of our manuscript for consideration at the Scientific Reports.

We appreciate the time and effort that you and the reviewers have devoted to providing your helpful feedback on our manuscript. We are grateful to you and the reviewers for your insightful and productive comments. The suggestions have made this a much better paper.

We have carefully considered and addressed the comments and integrated changes to indicate the comments raised by the reviewers. We have also extensively rewritten the text according to the reviewers' suggestion to make it less speculative and highlighted the changes within the manuscript. All the authors have equally contributed towards the revision of the manuscript. Please see below for a point-by-point response to the reviewers' comments and concerns. Our responses are shown in Blue italics, while reviewers' comments are in Black Bold italics. We look forward to your response and hope the revision will enable you to accept this version of the manuscript.

We look forward to hearing from you and thank you for your valuable and constructive comments.

Sincerely,

The Authors

Reviewer 2 Report

Dear Authors,

Thank you for your revision.

Comment 1: Although, I don't agree with your reasoning, I accept the editors decision. 

Comment 2: The revisions introduced indicate more clearly the scope of the paper, but the abstract still needs language correction. 

Comment 3: The introduction was barely modified. It still lacks a clearly defined purpose and motivation for the study. 

Comment 4: Although the process could have been described in more detail, the changes made are acceptable. 

Comment 5: The related work section was modified according to my comment. 

Comment 6: I can't find any changes in section 5.1. Maybe the wrong version of the manuscript was sent, as the cover letter says that this section was revised. 

In addition, proofreading is extremely important for this manuscript. 
There are a great many grammatical errors in the text that make the manuscript difficult to read. Especially the language problems occur in the first chapters, i. e. 1, 2 and 3. The article should be proofread by a native reader before publication. I'll give just two examples:
1. missing is: With music videos, this study not just analyzing sound, but visual information as well because visual information includes essential emotional clues such as facial expressions, gestures, physical reactions to environmental changes
2. Third form of the verb: First, multimodal representation work efficiently to capture all the acoustic and visual emotional clues included in the music video.

Author Response

(The authors gave the same response as above.)
